# Prediction of overt hepatic encephalopathy by the continuous reaction time method and the portosystemic encephalopathy syndrome test in clinically mentally unimpaired patients with cirrhosis

**Charlotte W. Wernberg**[1]*, **Ove B. Schaffalitzky de Muckadell**[2], **Hendrik Vilstrup**[3], **Mette M. Lauridsen**[1]

**1** Department of Gastroenterology, Hospital of Southwest Jutland, Esbjerg, Region of Southwest of Denmark, Denmark, **2** Department for Medical Gastrointestinal Diseases, Odense University Hospital, Odense, Region of Southwest of Denmark, Denmark, **3** Department of Hepatology and Gastroenterology, Aarhus University Hospital, Central Denmark Region, Denmark

* charlotte.wilhelmina.wernberg@rsyd.dk

## Abstract

### Background and aim

Predicting overt hepatic encephalopathy (OHE) is important because the condition is frequent, often requires hospitalization and is potentially preventable. The risk of OHE is related to pre-existing discrete cognitive defects, and for clinical practice it is recommended to apply two different psychometric tests to detect such deficits. We used the continuous reaction time test (CRT) and the portosystemic encephalopathy (PSE) syndrome test and examined their single and combined value for OHE prediction in cirrhosis patients.

### Patients and methods

We studied 130 clinically mentally unimpaired cirrhosis patients by the two tests and followed them for an average of 38.5 months. The CRT measures velocity and stability of motor reaction times to 150 repeated auditory signals. The PSE is a five sub-set paper-and-pencil test battery evaluating cognitive and psychomotor processing, speed and vision-motor coordination. We collected data on episodes of OHE during follow-up. The clinical course was analysed in patient groups according to the outcome of each test and of both tests together. No anti-HE treatment was initiated except for cases with OHE.

### Results

At baseline, the CRT test was abnormal in 74 patients and the PSE in 47. During follow-up 35 patients (27%) experienced 74 OHE events. 23 patients with abnormal CRT experienced OHE (prediction sensitivity 65%). The PSE predicted OHE in 14 patients (prediction

**Data Availability Statement:** All relevant data are within the manuscript and its Supporting

Information file. In order to secure all participants' privacy, identifying information (birthdate, age, gender, rare devices, admission- and death-dates) as well as some pseudo-identifying information has been removed. The Supporting information is therefore considered highly anonymous and there is no risk of potential identifiable material and we have therefore not obtained informed consent on open data from participants.

**Funding:** The authors received no specific funding for this work.

**Competing interests:** MM Lauridsen has given lectures paid by Norgine, MM Lauridsen has a consultant agreement with Umecrine Cognition, MM Lauridsen is on the steering committee in the International Society for Hepatic Encephalopathy and Nitrogen Metabolism (ISHEN). This does not alter our adherence to PLOS ONE policies on sharing data and materials.

sensitivity 40%). One or both tests were abnormal in 87/130 (67%) and this predicted OHE in 27 patients (21%) (prediction sensitivity 77%).

## Conclusion

The CRT test was clinically useful in identifying two-thirds of clinically mentally unimpaired cirrhosis patients who later experienced OHE, and the use of both the CRT and PSE showed satisfactory prediction by identifying three-fourths of later OHE cases.

## Introduction

Overt hepatic encephalopathy (OHE) requires treatment and is a frequent admission cause for patients with liver cirrhosis. Prognosticating OHE in order to prevent it is important because the condition impairs quality of life, often reoccurs, necessitates hospitalization, and is potentially preventable [1, 2].

It is known that the risk of developing OHE rises with decreasing liver function and portosystemic shunting [3]. Pre-existing discrete cognitive deficits caused by minimal hepatic encephalopathy (MHE), hyponatremia, and comorbidities are also associated with an increased risk of OHE [4–8]. The detection and quantification of such discrete cognitive deficits is done by psychometric testing [9]. International guidelines recommend patients with liver cirrhosis be cognitively evaluated by one or two validated psychometric tests meeting local preferences and resources [9]. A handful of tests are validated for the purpose. The continuous reaction time test (CRT) and the portosystemic encephalopathy (PSE) syndrome test are two such tests [9].

The CRT has been used for more than 3 decades, mostly in Scandinavia, for diagnosing and grading MHE and is validated regarding normal values, reproducibility, effects of age, gender, educational level, comorbid diseases, organic brain disease, sleep deprivation, and its ability to identify patients who will improve their CRT by anti-HE treatment [10–18]. In recent years the parallel use of CRT and the recommended surrogate gold standard test, the PSE, has gained momentum. The purpose of a two-test approach is to cover more cognitive domains and thus increase the collective test sensitivity [9, 19].

In spite of the availability of validated psychometric tests few centres routinely undertake psychometrics [20]. One important issue is that it is not clarified by the existing studies to what extent psychometric tests can predict OHE events. Such knowledge could motivate preventive anti-HE treatment and shed light on whether psychometric testing is worth the effort.

The aim of our study therefore is to examine the single and combined predictive value of the continuous reaction time test and the portosystemic encephalopathy syndrome test for OHE occurrence.

Our hypothesis is that each psychometric test has a good predictive value for future OHE and that their combined use offers the best predictive value. This assumption is based on the fact that a combined use detects a broader spectrum of cognitive deficits.

## Patients and methods

### Patients

The cohort consists of 130 patients with liver cirrhosis mainly of alcoholic aetiology (82%). The cohort was originally established to conduct a cross-sectional study, but with the approval

of the local ethical committee we followed the cohort with the aim of registering later episodes of OHE, other admission causes, and death (10).

Inclusion criterions were: liver cirrhosis diagnosed by liver biopsy or by classic combinations of clinical, biochemical, image technique criteria; and the absence of OHE by the clinical assessment of the patient's usual gastroenterologist.

Exclusion criteria were: HE West Haven Grade 1 or more, dementia (Mini Mental State Examination test score <24), prior cerebral insult, on-going alcohol use and acute intoxication at the day of inclusion, use of psychoactive medication (stable use of SSRI was allowed), severe hyponatraemia (p-Na < 125 mmol/L), renal failure (creatinine > 1.7 mg/dL), myxoedema, sepsis, or gastro-intestinal bleeding within a week before the tests. No one had diagnosed hepatocellular carcinoma at inclusion or at the time for psychometric testing. Inclusion and exclusion criteria and baseline data are previously published [12]. Regarding alcohol use we did not require a certain abstinence period but sobriety upon testing by simply asking patients if they had been drinking before commencing psychometric testing. Some patients had on/off alcohol use during follow up.

Patients were recruited from the departments of hepatology out-patient clinics on the Hospital of Southwest Jutland and University Hospital of Odense (Denmark) between February 2013 and November 2014. All patients had a CRT and a PSE test on the day of inclusion. The study duration was from inclusion to study termination in May 2017. We collected data on all hospital contacts including hospital admissions for OHE, other admission causes, and death. Patients remained in the study after the first recorded OHE episode i.e. we collected data on repeated OHE events. At inclusion, only subjects that earlier had been diagnosed with OHE received anti-HE treatment according to standard protocol with lactulose. An exception was 6/130 patients (5%) who received preventive anti-HE treatment for investigative purpose, they received a combination of lactulose, branched-chain amino acids and rifaximin for 3 months [10].

All patients gave their written informed consent. The study was approved by the Danish Data Protection Agency and The Local Committee on Health Research Ethics, and the follow up analysis presented here was reported to the regional data protection office, all according to Danish law.

## Methods

The CRT is a short, computerized test (equipment from EKHO, www.Bitmatic.com, Aarhus, Denmark). The test measures motor reaction speed, sustained attention, and inhibitory control, which are all key abilities of daily life functioning [14]. The test registers motor reaction time to 150 auditory stimuli (beeps at 500 Hz and 90 dB via headphones). The auditory stimuli are given at random intervals from 2 to 6 seconds and the patient is instructed to immediately press a handheld trigger button, in response to each beep. The software registers the response times and calculates the CRT index (the ratio: 50 percentile/ (90 minus 10 percentile)), which is the main test result and a measure of the patient's reaction time stability. A CRT index value below 1.9 is abnormal and may indicate MHE. The test has been in use for 3 decades in an increasing number of hospitals and is validated in the target population. Reaction time stability (CRT index) is not influenced by gender, age or intelligence; and is able to identify patients who improve cognition by anti-HE treatment [10, 14, 15]. The CRT index may be adversely affected in chronic medical conditions i.e. severe obstructive lung disease and heart failure [11].

The PSE test is a widely used test battery that can be completed in 20 minutes and often serves as a comparator test between centres and have good external validity [21]. The PSE consists of five paper-pencil tests: Digit symbol test (DST), number connection test A and B (NCT

A and B), serial dotting test (SDOT), and Line Tracing Test (LTT), which evaluate cognitive and psychomotor processing, speed and vision-motor coordination [19, 22]. The total score—portosystemic hepatic encephalopathy score (PHES)—ranges from -18 to 6 and summarizes the overall performance in the five tests relative to norm values. A result -5 or below is abnormal and may indicate MHE. Standard normative data of the PHES have been developed in German, Spanish and Italian [23]. We evaluated the test using the German normal values since Danish normal values are not yet available. There is no gender effect, and no significant effect of educational level. Four versions of the test battery are available to prevent the learning effect of repeated testing. The PSE test can be obtained from Hannover Medical School (Hannover, Germany), which holds the copyright (Weissenborn.karin@mh-hannover.de) [9].

**Data collection and management.** One observer reviewed the participants' electronic medical files between March and May 2017 and collected data on OHE admissions and death, and other adverse outcomes i.e. orthopaedic injury, falls, infections, ascites, variceal bleeds and dehydration/signs of acute kidney injury. The available electronic medical file system is used at all hospitals in the Region of Southern Denmark. All admissions and out-patients visits in the region are therefore registered. Death outside of the hospitals is registered as well. Hospitalization outside of the region or in another country, without this being mentioned in the patient file, is unlikely in Denmark. If the patient's admission was registered with more than one diagnosis, the main diagnosis was used. Whenever OHE was described in the discharge letters this was regarded as a study relevant diagnosis. OHE does not have a formal ICD-10 code, so the cases were sought out manually. The observer was blinded to the psychometric test results at inclusion. No subjects withdrew their consent, and none moved during study period. All were kept in the study after the first OHE event. Two were transplanted during follow up and 57 died and were withdrawn from our analysis from the time of transplantation/death.

**Statistical analysis.** Statistical analyses and graphics were performed with STATA (14.2 STATA Cooperation, College station, Texas, USA). Before analyses the patients were categorized according to their initial CRT and PSE test results into four test groups ("CRT and PSE normal", "CRT abnormal and PSE normal", "CRT normal and PSE abnormal", "CRT and PSE abnormal") to evaluate the predictive value for OHE of single and combined use of CRT and PSE tests.

Categorical variables were compared using the Fisher's exact test (due to few observations in some cells when using $X^2$); quantitative variables were compared using analysis of variance (ANOVA). Continuous variables that followed normal distribution were compared using student's t-test. To compare admissions with OHE we performed robust regression analysis, which was tested appropriate due to some test groups showed variance, heterogeneity and couldn't be considered normally distributed. All reported P-values are 2-tailed. P values $\leq$.05 were considered as statistically significant. Pairwise comparisons of significant p-values by the Tukey's HSD were made.

## Results

Baseline characteristics of the whole cohort, and the sub-groups categorized by psychometric test results, are presented in Table 1. The total follow-up time was 5,000 months and the median follow-up 38.5 months (range 27–48 months). Out of 130, 29 (22%) had had their first OHE episode before entering the study, but none had OHE at the time of inclusion. Thirty-five patients (27%) experienced at least one admission with OHE during follow up, and the total number of observed episodes was 74–24/35 patients (69%) had one episode of OHE during follow-up while 11 patients (31%) had more than one OHE episode. OHE was together with infections the most frequent liver-related cause for hospital contact, followed by ascites (Table 2).

**Table 1. Baseline characteristics.**

| | Overall cohort (n = 130) | CRT and PSE normal (n = 43) | CRT abnormal and PSE normal (n = 40) | CRT normal and PSE abnormal (n = 13) | CRT and PSE abnormal (n = 34) | P-value ANOVA |
|---|---|---|---|---|---|---|
| CRT index, mean (SE) | 1.9 (0.06) | 2.6 (0.11) | 1.5 (0.05) | 2.3 (0.08) | 1.4 (0.05) | **<0.001$^q$** |
| PHES, mean (SE) | -3.9 (0.40) | -0.9 (0.32) | -1.4 (0.37) | -8.2 (0.85) | -9.1 (0.52) | **<0.001$^x$** |
| Male gender % (n) | 67.7 (88) | 72 (31) | 55 (22) | 92 (12) | 68 (23) | 0.07 |
| Age, median years (range) | 59 (40–79) | 57 (44–76) | 59 (40–79) | 65 (49–72) | 62 (40–75) | 0.25 |
| MELD score, median (range) | 10 (6–29) | 10 (6–25) | 12.5 (6–29) | 10 (7–21) | 13 (6–21) | 0.24 |
| Child-Pugh score, median (range) | 6 (3–12) | 6 (3–12) | 6 (5–12) | 6 (5–11) | 7.5 (5–12) | **<0.03$^y$** |
| Child-Pugh A % (n) | 55.4 (72) | 69.8 (30) | 52.5 (21) | 53.9 (7) | 41.2 (14) | 0.09 |
| Child-Pugh B % (n) | 33.1 (43) | 20.9 (9) | 40.0 (16) | 38.5 (5) | 38.2 (13) | 0.23 |
| Child-Pugh C % (n) | 11.5 (15) | 9.3 (4) | 7.5 (3) | 7.7 (1) | 20.6 (7) | 0.29 |
| Previous HE % (n) | 22.3 (29) | 20.9 (9) | 17.5 (7) | 30.7 (4) | 26.5 (9) | 0.70 |
| Ascites % (n) | 48.5 (63) | 34.9 (15) | 55.0 (22) | 46.2 (6) | 58.8 (20) | 0.15 |
| TIPS % (n) | 5.4 (7) | 2.3 (1) | 0.0 (0) | 23.1 (3) | 8.8 (3) | **<0.017$^z$** |
| Varices % (n) | 59.2 (77) | 65.1 (28) | 60.0 (24) | 53.9 (7) | 53.0 (18) | 0.73 |
| Education, years (SD) | 10.9 (2.6) | 11.2 (2.6) | 11.6 (2.8) | 10.2 (2.7) | 9.9 (1.8) | **0.03 $^\Delta$** |
| Charlson comorbidity index | 3.5 (1.2) | 3.3 (1.0) | 3.3 (1.4) | 3.5 (0.8) | 3.8 (1.1) | 0.30 |

130 patients with liver cirrhosis, tested for the presence of discrete cognitive deficits using the continuous reaction times test (CRT) and portosystemic encephalopathy (PSE) syndrome test. Data are expressed as median (range), unless specified otherwise. P-values in bold show significant outcome at a level below 0.05

Abbreviations: CRT: continuous reaction time test (CRT index abnormal if below 1.9), PSE: portosystemic encephalopathy syndrome test (PHES psychometric hepatic encephalopathy score, abnormal if below -4), MELD: model for end-stage liver disease, HE: hepatic encephalopathy, TIPS: trans jugular intrahepatic portosystemic shunt.

Statistical differences (P ≤ .05) between groups indicated as comparison between:

q: CRT and PSE normal vs. CRT abnormal and PSE normal; CRT abnormal and PSE normal vs. CRT normal and PSE abnormal; CRT and PSE normal vs. CRT and PSE abnormal; CRT normal and PSE abnormal vs. CRT and PSE abnormal

X: CRT abnormal and PSE normal vs. CRT normal and PSE abnormal; CRT and PSE normal vs. CRT normal and PSE abnormal; CRT and PSE normal vs. CRT and PSE abnormal; CRT abnormal and PSE normal vs. CRT and PSE abnormal

Y: CRT and PSE normal vs. CRT and PSE abnormal

Z: CRT and PSE normal vs. CRT normal and PSE abnormal; CRT abnormal and PSE normal vs. CRT normal and PSE abnormal

Δ: CRT abnormal and PSE normal vs. CRT and PSE abnormal

## Predicting OHE using only the CRT test

Seventy-four out of the 130 patients (57%) exhibited an abnormal CRT and 23 of these (31%) experienced at least one episode of OHE—prediction sensitivity 65% (Table 2 and Fig 1). These 23 patients (31%) were responsible for 73% of all OHE admissions of the total cohort, reflecting that several of them experienced more than one OHE episode in spite of being discharged with preventive anti-HE treatment according to guideline.

## Predicting OHE using only the PSE test

Forty-seven of the 130 patients (36%) had an abnormal PHES and 14 of these (30%) developed at least one episode of OHE, yielding a prediction sensitivity of 40% (Table 2 and Fig 1). These 14 patients (30%) were responsible for 42% of all OHE admissions of the total cohort.

**Table 2. Events requiring admissions during follow-up.**

| | Overall cohort (n = 130 | CRT and PSE normal (n = 43) | CRT abnormal and PSE normal (n = 40) | CRT normal and PSE abnormal (n = 12) | CRT and PSE abnormal (n = 34) | P-value ANOVA |
|---|---|---|---|---|---|---|
| **Cirrhosis related admissions** | | | | | | |
| *OHE % (n)* | 27 (35) | 19 (8) | 32 (13) | 31 (4) | 29 (10) | 0.57 |
| *Number of events* | 74 | 14 | 29 | 6 | 25 | |
| *Ascites % (n)* | 21 (27) | 19 (8) | 28 (11) | 15 (2) | 18 (6) | 0.19 |
| *Number of events* | 70 | 15 | 31 | 15 | 9 | |
| *Infection % (n)* | 28 (37) | 33 (14) | 25 (10) | 0 (0) | 35 (12) | 0.15 |
| *Number of events* | 57 | 23 | 14 | 0 | 14 | |
| *Variceal bleeding % (n)* | 12 (16) | 21 (9) | 10 (4) | 15 (2) | 3 (1) | 0.10 |
| *Number of events* | 33 | 21 | 8 | 3 | 1 | |
| *Alcohol related % (n)* | 8 (10) | 5 (2) | 10 (4) | 8 (1) | 9 (3) | 0.45 |
| *Number of events* | 21 | 3 | 12 | 2 | 4 | |
| **Other frequent admission causes** | | | | | | |
| **Orthopaedic*** % (n) | 18 (23) | 14 (6) | 18 (7) | 15 (2) | 24 (8) | 0.51 |
| *Number of events* | 31 | 6 | 11 | 3 | 11 | |
| **Dehydration % (n)** | 9 (12) | 5 (2) | 13 (5) | 0 (0) | 12 (4) | 0.26 |
| *Number of events* | 22 | 2 | 9 | 0 | 5 | |
| **Death** | 44 (57) | 42 (18) | 40 (16) | 31 (4) | 56 (19) | 0.21 |

In a cohort of 130 patients with liver cirrhosis, tested for the presence of discrete cognitive deficits using CRT and PSE test. Data are expressed as percentage of number of patients in each test group, Specific events or admission causes also has the total number of events specified under each group. P-values in bold show significant outcome at a level below 0.05. The reference value used in the ANOVA analysis were the group CRT and PSE normal.

*Acute orthopaedic injuries, fractures and falls

OHE: overt hepatic encephalopathy.

## Predictive value of the combined use of CRT and PSE test

When combining the two tests, *one or both tests were abnormal* in 87/130 (67%) patients, and of these 27 patients (31%) developed at least one OHE episode, resulting in a prediction sensitivity of 77% (Table 2 and Fig 1). These 27 patients (31%) were responsible for 81% of OHE-admissions (Table 2).

Thirty-four out of 130 patients (26%) had *abnormal results of both tests* and of these 10 (29%) experienced at least on OHE episode. This gives a low prediction sensitivity of 29% (Fig 1). These 10 patients (29%) were responsible for only 34% of OHE admissions.

Eight of 43 patients (19%) with two normal tests experienced OHE and were responsible for 19% of OHE admissions (Table 2).

## OHE admission patterns

Sixty-eight per cent (24/35) of patients experienced their first and only OHE admission during follow up. In the remaining 31% (11/35) OHE was relapsing. Most subjects relapsed 2–5 times, but two patients (2/35, 6%) experienced more relapses ascribed to uncontrolled recurring infections. There was a tendency that baseline CRT and PSE results were worse with increasing number of OHE events during follow up (S1 Table). Among the subjects with previous OHE at inclusion, 69% did not have another episode during follow up. For the OHE naïve subjects 74% still did not experience an episode during follow-up. For all test groups admission rates

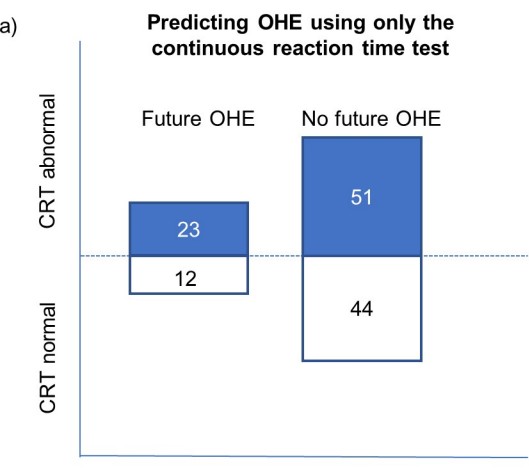

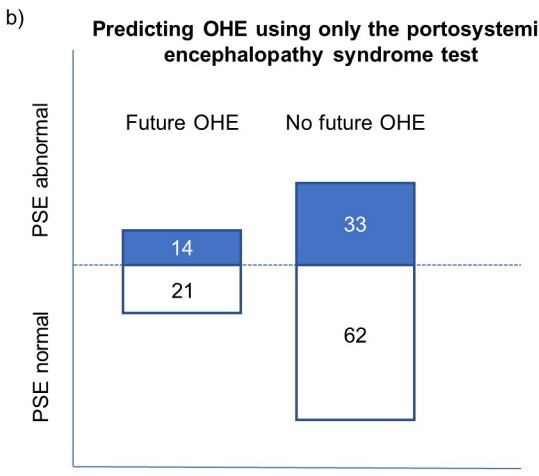

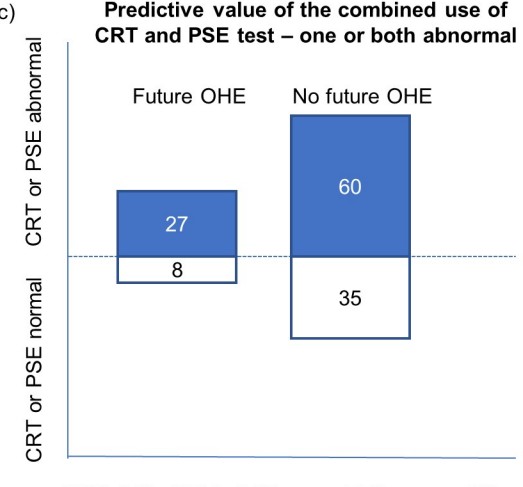

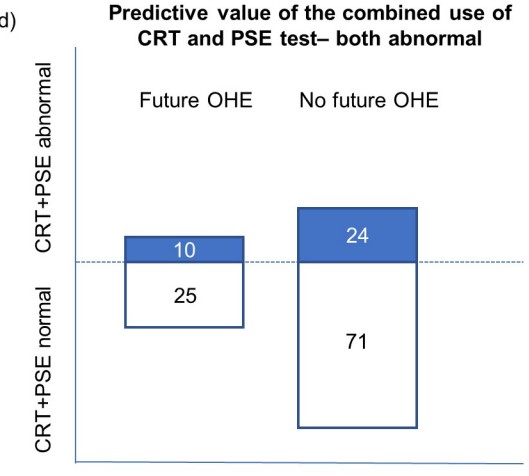

**Fig 1. Four scenarios for predicting OHE; with single or combined use of the two psychometric tests used.** All numbers inside the boxes (blue or white) represent the number of patients, having an abnormal (*blue*) or normal (*white*) test result. Panel a): A scenario with the single use of the continuous reaction time test (CRT), Panel b): Scenario with the single use of the PSE test. Panel c): Combined use where either abnormal test is diagnostic, Panel d): Combined use where two abnormal tests are diagnostic. *PPV = positive predictive value; NPV = negative predictive value; Sens = sensitivity; Spec = specificity.*

were higher among the groups with abnormal test results and highest in the group where CRT and PSE both were abnormal. (Fig 2). However, by multivariate analysis including potential known confounders (age, gender, MELD-score, Child-Pugh points), the higher admission rates in the groups with abnormal test results seen in the graphs (Fig 2) were not statistically significant (S2 Table).

## Discussion

The aim of this study was to examine the predictive value of the CRT and the PSE tests for OHE occurrence in a well-defined cohort of patients with liver cirrhosis. We assessed the predictive value of both single and combined use of the two tests for an average of 38.5 months

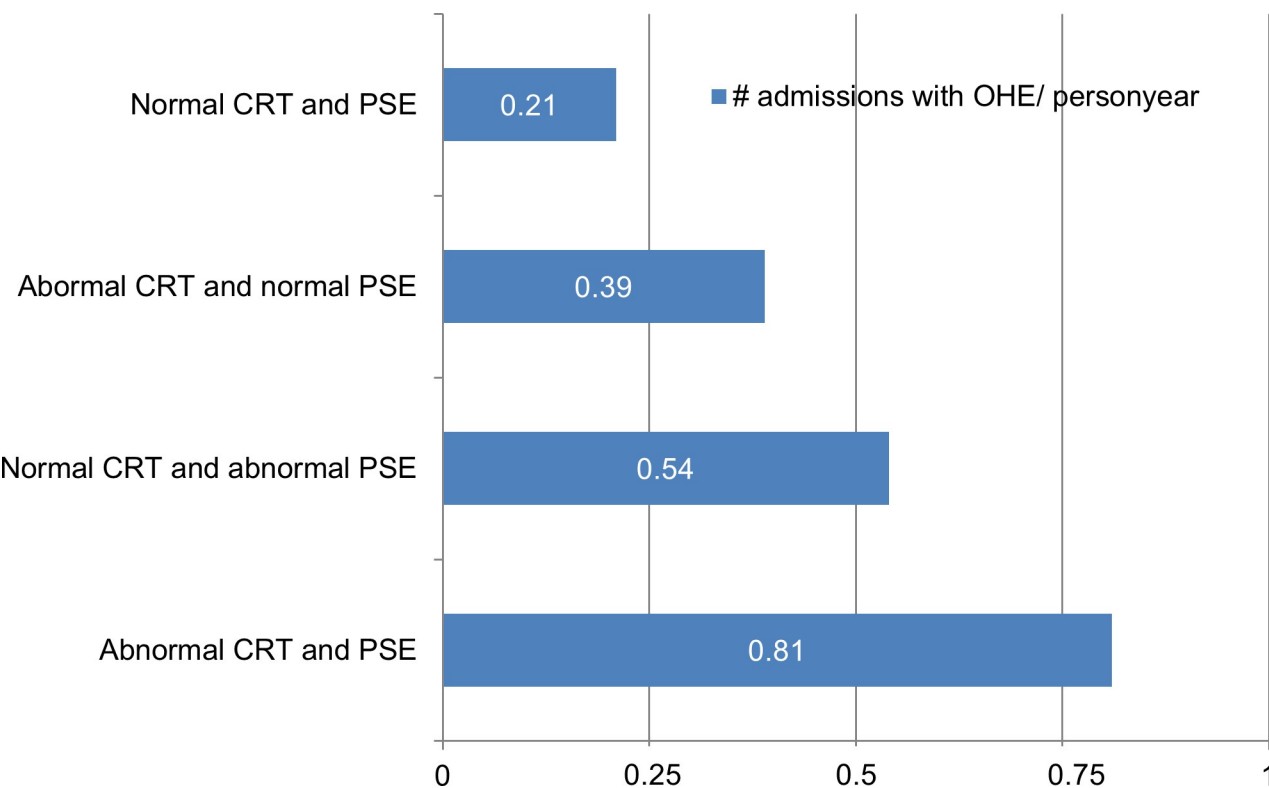

**Fig 2. The number of admissions with overt hepatic encephalopathy (OHE), per person year, in the four test groups.** In the group CRT and PSE normal (0.21), CRT abnormal and PSE normal (0.39), CRT normal and PSE abnormal (0.54) and CRT and PSE abnormal (0.81). Results are not statistically significantly different between groups after controlling for known confounders.

follow up. Our hypothesis was that their combined use would offer the best predictive value since such an approach detects a broader spectrum of cognitive deficits i.e. the patients who are most cognitively impaired. Prediction was confirmed to be relevant, by 27% of the tested patients later developing OHE and thus being important to identify as prospective candidates for preventive anti-HE treatment.

We found that the CRT method alone predicted 65% of later OHE episodes, the PHES alone 40%, and both together 77%.

For the use of any given predictive test the desirable level of predictive value is a decisive issue. Prediction of clinical events is never perfect. The desirable level varies with the specific clinical problem and is determined by a number of factors such as the frequency of the problem, if it is predicted by other factors, how serious it is, how complicated the test is, and the efficacy, cost and safety of the preventive measure. OHE, as noted above, is a frequent complication for mentally unimpaired cirrhosis patients; it was not satisfactory predicted by clinical data including prior HE episodes; it is a very serious complication; the psychometric tests are non-invasive, fast, and cheap [9]; OHE is preventable by lactulose treatment with a low number needed to be treated (NNT = 4), and lactulose is widely available, cheap, safe, and has negligible side effects [24]. These statements, although not giving a definite level, suggest that prediction of half of the later OHE cases by psychometry is meaningful, of two thirds useful, and of three fourths satisfactory.

According to such levels, the CRT method alone was useful (predicted 65%), the PHES alone not sufficient (predicted 40%), and their combined use gave satisfactory (predicted 77%).

A limitation of our study is that we primarily dealt with prediction of OHE occurrence because OHE is serious, not satisfactory predicted by clinical data and preventable. Prediction of non-occurrence might have been of interest if the preventive measure was expensive or had significant side effects. In that case, a high negative predictive value and a low number of false positives—a high specificity—would be of importance. The negative predictive value of the CRT alone was 78%, of the PHES 75%, and of the combined use 81%. These numbers suggest that the CRT and PSE, alone and combined, are fairly good at ruling out future OHE.

Further our study is limited by the fact that we did not systematically control for participants on/off alcohol consumption (we simply asked them if they had been drinking before commencing psychometric testing) or adherence/non-adherence to any anti-HE treatments. Also, the use of German PSE normal values may not me optimal although Germans and Danes hardly differ significantly.

Other recent studies [25–28] have dealt with prediction of OHE and one of them used a similar approach to ours [27] by evaluating single versus combined testing. A major difference from our approach is however that the study by Duarte-Rojo et. al [27] required both psychometric tests to be abnormal to give the diagnosis of MHE and accordingly found lower predictive values in the range 30–55%. Similarly, we found that in patients with abnormal CRT and PSE prediction sensitivity was only 29%. The reason for that is likely that by requiring two psychometric tests to be abnormal we overlook all the patients whose cognitive deficits are demonstrated in just one test but who still have an elevated risk for OHE as demonstrated by our findings. These results show very clearly that not all HE patients are affected by HE to the same degree and in the same cognitive functions and illustrate why sensitivity is increased by a combined use of tests requiring just one of them to be abnormal.

Duarte-Rojo et. al [27] found that the single use of the PSE test had a predictive value of 36% which is similar to our findings (40% PHES predictive value). Accordingly, both studies suggest that the predictive value of the PSE test is outperformed by that of the Stroop EncephalApp (55%) and the CRT test (65%).

## Conclusion

The continuous reaction time test was clinically useful in identifying two-thirds of clinically mentally unimpaired cirrhosis patients who later experienced over hepatic encephalopathy, and the combined use of the continuous reaction time test and the portosystemic encephalopathy syndrome test showed satisfactory prediction by identifying three-fourths of later overt hepatic encephalopathy cases. We suggest our findings to imply that all patients with liver cirrhosis should be psychometrically tested and those with abnormal continuous reaction time test alone and abnormal continuous reaction time test and portosystemic encephalopathy syndrome test together; should be offered preventive lactulose treatment and close follow up regarding the effect on cognition.

## Supporting information

**S1 Data. Rawdata output on the cohort of 130 patients with cirrhosis.** Data is anonymisized: gender, social security number, age and some other characteristics is deleted so that rare events will still not be traceable back to each subject.
(XLSX)

**S1 Fig. Number of admissions with overt hepatic encephalopathy (OHE), per person year, in subjects with normal or abnormal CRTindex.** Disregarding psychometric hepatic

encephalopathy score.
(TIF)

**S2 Fig. Number of admissions with overt hepatic encephalopathy (OHE), per person year, in subjects with normal or abnormal PSE.** Disregarding CRT index.
(TIF)

**S1 Table. Psychometric test results in patients with: No OHE event, a single OHE event and recurring OHE events.**
(DOCX)

**S2 Table. Analysis of association: Multivariable linear regression analysis on OHE admission per person year, coefficients and confidence intervals.** Supplementary statistical output—for linear logistic regression model with margins. Supplementary information to Fig 2.
(DOCX)

## Author Contributions

**Conceptualization:** Mette M. Lauridsen.

**Formal analysis:** Charlotte W. Wernberg.

**Investigation:** Mette M. Lauridsen.

**Methodology:** Charlotte W. Wernberg, Mette M. Lauridsen.

**Project administration:** Charlotte W. Wernberg, Mette M. Lauridsen.

**Resources:** Ove B. Schaffalitzky de Muckadell, Hendrik Vilstrup.

**Software:** Charlotte W. Wernberg.

**Supervision:** Hendrik Vilstrup, Mette M. Lauridsen.

**Writing – original draft:** Charlotte W. Wernberg, Mette M. Lauridsen.

**Writing – review & editing:** Charlotte W. Wernberg, Ove B. Schaffalitzky de Muckadell, Hendrik Vilstrup, Mette M. Lauridsen.

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
