## [Decision Letter · Decision Letter 0]

4 Oct 2019

PONE-D-19-25609

Prediction of overt hepatic encephalopathy by the continuous reaction time method and the portosystemic encephalopathy syndrome test in clinically mentally unimpaired patients with cirrhosis

PLOS ONE

Dear Ms Wernberg,

Thank you for submitting your manuscript to PLOS ONE. After careful consideration, we feel that it has merit but does not fully meet PLOS ONE’s publication criteria as it currently stands. Therefore, we invite you to submit a revised version of the manuscript that addresses the points raised during the review process.

As you can see, both reviewers evaluated your study as interesting and carefully performed, however, suggested additional information to be added/additional analyses to be performed.

We would appreciate receiving your revised manuscript by Nov 18 2019 11:59PM. To enhance the reproducibility of your results, we recommend that if applicable you deposit your laboratory protocols in protocols.io, where a protocol can be assigned its own identifier (DOI) such that it can be cited independently in the future. For instructions see: http://journals.plos.org/plosone/s/submission-guidelines#loc-laboratory-protocols

We look forward to receiving your revised manuscript.

Kind regards,

Pavel Strnad

Academic Editor

PLOS ONE

**Journal Requirements:**

2. In the Methods, please cite the original references describing the development of the PSE and CRT tests, or describe how others may gain access to them. If the PSE test was published under a copyright no more restrictive than CC-BY, please include a copy as Supporting Information.

3.  In the ethics statement in the Methods and online submission information, please ensure that you have specified whether consent was informed."

**Comments to the Author**

1. Is the manuscript technically sound, and do the data support the conclusions?

Reviewer #1: Yes

Reviewer #2: Yes

2. Has the statistical analysis been performed appropriately and rigorously? 

Reviewer #1: Yes

Reviewer #2: Yes

3. Have the authors made all data underlying the findings in their manuscript fully available?

Reviewer #1: Yes

Reviewer #2: Yes

4. Is the manuscript presented in an intelligible fashion and written in standard English?

Reviewer #1: Yes

Reviewer #2: Yes

5. Review Comments to the Author

Reviewer #1: Wernberg et al. investigated the predictive ability of CRT and PHES regarding the development of OHE in patients with liver cirrhosis. This study represents a lot of hard work that is carefully done. I have only a few comments:

1. Since medical files were reviewed retrospectively, how did you exclude that patients were hospitalized with OHE in different hospitals?

2. For how long did patients have to stop drinking before being included into the study?

3. Did you include patients with HCC?

4. Is it correct that no patient was lost to follow-up?

5. Is it correct that no patient required liver transplantation?

6. Currently, you are not dealing with competing events like death or liver transplantation in your analyses. I would recommend adding cumulative incidence plots for every testing strategy. Here death and liver transplantation can be considered as competing events for OHE. Additionally, please provide a table with the hazard ratios and confidence intervals of your multivariable analyses.

7. Some studies already investigated the predictive ability of PHES regarding development of OHE. Additionally, a recent study by Duarte-Rojo et al. tested the utility of single versus combined testing (Duarte-Rojo et al, Metab Brain Dis 2019). Here, the authors conclude that combined testing is not superior to single testing to predict OHE. This is somehow in contrast to your conclusion. Can the authors comment on this?

8. Line 25: “The negative predictive value of the CRT alone was 79%”. In figure 1 it is 78%. Please correct this.

9. Line 249:”of three fourths excellent”; Line 262 “showed excellent prediction”. A sensitivity of three fourths cannot be declared as excellent. Please tone these sentences down.

Reviewer #2: In this manuscript Wernberg et al. assessed the predictive value of the continuous reaction time method (CRT) and the portosystemic encephalopathy syndrome (PSE) test for episodes of overt hepatic encephalopathy in patients with liver cirrhosis. They assessed 130 patients with liver cirrhosis with both tests and followed them for an average of 38.5 months. Of the 74 patients with an abnormal CRT 23 patients developed an overt HE and of the 47 patients with an abnormal PSE test 14 patients developed overt HE. When one or both tests were abnormal the prediction sensitivity was 77%. The authors conclude that the CRT alone but especially the combination of CRT and PSE test are clinically useful to identify patients who are at risk to develop overt HE. The authors suggest that all patients with liver cirrhosis should be tested psychometrically and if pathological, preventive treatment should be initiated.

This manuscript addresses an important topic and I congratulate the authors for performing this follow-up study. Testing patients with liver cirrhosis psychometrically is important and has clinical impact.

Comments:

In general:

- The authors should provide references for statements where applicable.

- Please add “%” after all patient numbers throughout the manuscript where applicable.

Abstract:

- Please revise the first sentence. I suggest that the authors delete the word “unpleasant” and replace it with “impairment of health related quality of life” and “cognitive impairment”.

- Please correct “Porto Systemic Encephalopathy Test (PSE)” to “portosystemic encephalopathy (PSE) syndrome test” and use this spelling throughout the manuscript.

- Please provide a hypothesis.

- Lines 30/31: The description of the PSE is wrong, please correct it. (The PSE does not only measure time, e.g. digit symbol test, line tracing test)

Introduction:

- Line 47: Please revise as described above

- Lines 52-55: Please revise this sentence.

- Please provide a hypothesis at the end of the introduction

Patients and methods:

- Line 78: Please define “no clinically detectable cognitive deficit”. Did the authors use a test?

- The inclusion and exclusion criteria should be described

- Lines 89/90: This sentence is hard to understand, please revise it.

- Lines 92-97: move this paragraph further up. Delete “which none met at the time of inclusion” as it is self-explanatory.

- Line 101: spelling error

- Line 107: please revise this sentence

- Line 115: I am surprised that the CRT is not influenced by age as it is well known that age influences reaction time in general. Can the authors please specify their comment?

- A reference for the PSE test is missing

- Please correct the description of the PSE test

- Why did the authors use norm data of the German population and not those of the Danish population?

- Lines 146/147: Could you please make the names of the different groups easier to understand e.g. “CRT/PSE normal”, “CRT abnormal/PSE normal” etc.

Results:

- Table 1: Can you please provide the pairwise analysis for the significant p-values

- Line 178: The title of table 2 is misunderstanding. Had all patients 38 months follow up?

- Lines 186/187/201/205: Please choose a clearer title. For example “Prediction of OHE using only CRT test” etc.

- Please provide the measured mean values of the CRT and PHES.

Discussion:

- I recommend to provide a short introduction at the beginning of the discussion

- Lines 234-237: Please revise this paragraph as it is hard to understand

- Line 246: Reference is missing

- Line 250: Please discuss the results of the paper in more detail. Especially why the PHES alone was “not meaningful”. This result contradicts that of other papers and should be discussed.

- Line 253: please describe the problem

- Did patients with recurrent episodes of OHE have worse psychometric test results than those with only one episode?

- The authors should list limitations of the study.

- In general: Please critically discuss the results of this study in detail and place them in the context of previous literature.

Conclusions:

- Lines 262/263: Please revise this sentence.

- The last sentence (line 265) contradicts the results and should be revised: “in those with two abnormal tests…” In the results section these patients only have a predictive value of 29%.

6. PLOS authors have the option to publish the peer review history of their article (what does this mean?). If published, this will include your full peer review and any attached files.

Reviewer #1: Yes: Dr. Christian Labenz

Reviewer #2: No

---

## [Author Response · Author response to Decision Letter 0]

17 Nov 2019

Response to reviewers 

Dear reviewers,

Thank you very much for your hard work; all comments and feedback! You have both made an effort and we have tried to address your comments to the best of our abilities and hope you feel the manuscript have improved. 

Please see attached file: Response to reviewers. 

The best of regards,

Charlotte Wernberg et al.

Text below is copy-pasted from the attached file: Response to reviewers. Format is therefore gone.

As to the journal comments:

1. In the Methods, please cite the original references describing the development of the PSE and CRT tests, or describe how others may gain access to them. If the PSE test was published under a copyright no more restrictive than CC-BY, please include a copy as Supporting Information. 

We have added the appropriate references.

2. In the ethics statement in the Methods and online submission information, please ensure that you have specified whether consent was informed." 

We have specified in the manuscript that informed written consent was achieved from all participants and the collection of follow up data was approved by the regional data protection agency.

Reviewer #1: 

1. Since medical files were reviewed retrospectively, how did you exclude that patients were hospitalized with OHE in different hospitals?

This is a very relevant point. The electronic medical file system was during the period used at all hospitals in the Region of Southern Denmark. All admissions and out-patients visits in the region are registered. Death outside of the hospitals is registered here as well. Hospitalization outside of the region, without this being mentioned in the patient file, is unlikely and a rare event. This has been clarified in the new manuscript page 7 lines 147-151.

2. For how long did patients have to stop drinking before being included into the study?

Ongoing alcohol abuse was an exclusion criterion, but we had no requirements as to the duration of abstinence. They were sober at the time of testing. Most patients had alcoholic cirrhosis and although they stated that they had stopped drinking at the time of inclusion into the cross-sectional cohort some of them had an on-off alcohol use during follow up. This has been mentioned in the manuscript page 5 line 90-93.

3. Did you include patients with HCC?

 None had diagnosed hepatocellular carcinoma at inclusion or at the time of psychometric testing. This has been clarified in the new manuscript page 5 line 89-90. 

4. Is it correct that no patient was lost to follow-up? 

Thank you very much for bringing this up. We were in fact able to account for all patients in this cohort during follow up and only those who died or were transplanted (cf. below) were censured. We kept patients in the study after their first OHE episode. 

5. Is it correct that no patient required liver transplantation?

Actually, two patients were transplanted. One of the tem died 4 months after due to complications to transplant. We had forgotten to exclude these persons in the admission causes analysis and admission causes statistics are now updated and new calculations are made.

6. Currently, you are not dealing with competing events like death or liver transplantation in your analyses. I would recommend adding cumulative incidence plots for every testing strategy. Here death and liver transplantation can be considered as competing events for OHE. Additionally, please provide a table with the hazard ratios and confidence intervals of your multivariable analyses.

Thank you for your comments and suggestions. We assume that you are referring to the analysis on admissions with OHE? We chose to split up the response here, as we read the comment as two separate issues. We hope we have addressed the issues accordingly and that we have interpreted the comments correctly. 

With regards to “…Currently, you are not dealing with competing events like death or liver transplantation in your analyses. I would recommend adding cumulative incidence plots for every testing strategy. …”. 

We have discussed this at an early phase with a statistician, before submitting the manuscript. We were advised against using classical survival analysis as seen in a cumulative incidence plot. This because some patients had several events with OHE and these needed to be treated as multiple failure-time data. We have therefore chosen not to use cumulative incidence plots for events with OHE in this manuscript.

With regards to “…Additionally, please provide a table with the hazard ratios and confidence intervals of your multivariable analyses…”

We have added a supplementary table with effect coefficients and 95%CI for the factors in the regression analysis and margins. There are however no hazard ratios when working with logistic regression. We have added a supplementary figure illustrating the admissions rates for a test strategy using only CRT and only PSE tests. 

7. Some studies already investigated the predictive ability of PHES regarding development of OHE. Additionally, a recent study by Duarte-Rojo et al. tested the utility of single versus combined testing (Duarte-Rojo et al, Metab Brain Dis 2019). Here, the authors conclude that combined testing is not superior to single testing to predict OHE. This is somehow in contrast to your conclusion. Can the authors comment on this?

The Duarte-Rojo study is very interesting. We have focused on the comparison with the Duarte-Rojo study in our discussion lines 305-320. 

8. Line 25: “The negative predictive value of the CRT alone was 79%”. In figure 1 it is 78%. Please correct this. 

Thank you for noticing, typo corrected in text. 

9. Line 249:”of three fourths excellent”; Line 262 “showed excellent prediction”. A sensitivity of three fourths cannot be declared as excellent. Please tone these sentences down. 

We agree on this, and have toned it down. 

Reviewer #2: 

In general:

- The authors should provide references for statements where applicable.

Thank you, we agree that there were too few references. This has now been changed for the better, with relevant and accurate references. We hope you agree.

- Please add “%” after all patient numbers throughout the manuscript where applicable. 

Thank you for noticing and make us aware of this, changes has been added all the way through. 

Abstract:

- Please revise the first sentence. I suggest that the authors delete the word “unpleasant” and replace it with “impairment of health related quality of life” and “cognitive impairment”. 

These suggestions improve the nuances to the text further, thank you. 

- Please correct “Porto Systemic Encephalopathy Test (PSE)” to “portosystemic encephalopathy (PSE) syndrome test” and use this spelling throughout the manuscript. 

Again, thank you very much. Spelling throughout the manuscript is corrected.

- Please provide a hypothesis.

Our hypothesis has been stated in the last section of the introduction. 

- Lines 30/31: The description of the PSE is wrong, please correct it. (The PSE does not only measure time, e.g. digit symbol test, line tracing test). 

We have changed the short description of PSE in the abstract:: The PSE is a five sub-set paper-and-pencil test battery evaluating cognitive and psychomotor processing, speed and vision-motor coordination.

And in the “methods” section: The total score — portosystemic hepatic encephalopathy score (PHES)— ranges from -18 to 6 and summarizes the overall performance in the five tests relative to norm values.

Introduction:

- Line 47: Please revise as described above. 

This has been revised; please see manuscript, lines 47-49.

- Lines 52-55: Please revise this sentence. 

The passage has been revised to “It is known that the risk of developing OHE rises with decreasing liver function and portosystemic shunting. Pre-existing discrete cognitive deficit caused by minimal hepatic encephalopathy (MHE), hyponatremia, and comorbidities are also associated with an increased risk of OHE. The detection and quantification of such discrete cognitive deficits is done by psychometric testing.”

- Please provide a hypothesis at the end of the introduction

Our hypothesis has been stated in the last section of the introduction.

Patients and methods:

- Line 78: Please define “no clinically detectable cognitive deficit”. Did the authors use a test? 

“no clinically detectable cognitive deficit” simply means no overt HE i.e. HE that can be detected clinically without the use of psychometric tests. This has been clarified in the exclusion criterions cf. below. 

- The inclusion and exclusion criteria should be described. 

We agree. These have been added. 

- Lines 89/90: This sentence is hard to understand, please revise it. 

Thank you, this has been revised. 

- Lines 92-97: move this paragraph further up. Delete “which none met at the time of inclusion” as it is self-explanatory. 

Paragraph is moved to a position further up in the paragraph.

- Line 101: spelling error. 

This has been corrected.

- Line 107: please revise this sentence. 

This has been done.

- Line 115: I am surprised that the CRT is not influenced by age as it is well known that age influences reaction time in general. Can the authors please specify their comment? 

The crude reaction times are very influenced by age and gender but he CRT measures reactions time stability as given by the CRT index and the reaction time stability is not influenced by age. It is however influenced by other comorbid diseases such as heart failure, diabetes and COPD. 

- A reference for the PSE test is missing. 

Reference is added. 

- Please correct the description of the PSE test. 

This has been done.

- Why did the authors use norm data of the German population and not those of the Danish population? 

There is no norm data in Danish yet. These are under development. 

- Lines 146/147: Could you please make the names of the different groups easier to understand e.g. “CRT/PSE normal”, “CRT abnormal/PSE normal” etc. 

We welcome the purposed names and will change the groups for these more logical and easier names, thank you!

Results:

- Table 1: Can you please provide the pairwise analysis for the significant p-values

We have now added pairwise analysis for the significant p-values to Table 1and hope that the way we have presented it is logical and satisfying. 

- Line 178: The title of table 2 is misunderstanding. Had all patients 38 months follow up?

“38 months” has been deleted since not all participants have 38 months of follow up time. Thank you for that comment. 

- Lines 186/187/201/205: Please choose a clearer title. For example “Prediction of OHE using only CRT test” etc.

Good idea. This has been corrected. 

- Please provide the measured mean values of the CRT and PHES.

This has been added in Table 1.

Discussion:

- I recommend to provide a short introduction at the beginning of the discussion

This has been added. 

- Lines 234-237: Please revise this paragraph as it is hard to understand

The paragraph has been divided and altered

- Line 246: Reference is missing 

Reference is added. 

- Line 250: Please discuss the results of the paper in more detail. Especially why the PHES alone was “not meaningful”. This result contradicts that of other papers and should be discussed.

We have changed the phrasing of this section and discussed results from other papers. 

- Line 253: please describe the problem

This has been done: This work primarily dealt with prediction of OHE occurrence because OHE is serious, not satisfactory predicted by clinical data including prior HE episodes and preventable.

- Did patients with recurrent episodes of OHE have worse psychometric test results than those with only one episode? 

Thank you for raising this question. We have chosen to add this data in the text in the results section, paragraph OHE admission patterns. There does not seem to be significant differences in the values of CRT index and PHES when we look at no/ one/ or >1 event with OHE. 

- The authors should list limitations of the study.

These have been added. 

- In general: Please critically discuss the results of this study in detail and place them in the context of previous literature.

We have chosen to discuss our findings in relation to the recent and very similar study by Duarte-Rojo as suggested by reviewer #2 and rewritten the discussion to a large extent. 

Conclusions:

- Lines 262/263: Please revise this sentence.

This has been done regarding the term “excellent” which has been changed to “satisfactory”. 

- The last sentence (line 265) contradicts the results and should be revised: “in those with two abnormal tests…” In the results section these patients only have a predictive value of 29%. 

Thanks for pointing this out. The sentence has been altered.

---

## [Decision Letter · Decision Letter 1]

25 Nov 2019

Prediction of overt hepatic encephalopathy by the continuous reaction time method and the portosystemic encephalopathy syndrome test in clinically mentally unimpaired patients with cirrhosis

PONE-D-19-25609R1

Dear Dr. Wernberg,

We are pleased to inform you that your manuscript has been judged scientifically suitable for publication and will be formally accepted for publication once it complies with all outstanding technical requirements.

With kind regards,

Pavel Strnad

Academic Editor

PLOS ONE

Additional Editor Comments (optional):

Reviewers' comments:

Reviewer's Responses to Questions

**Comments to the Author**

1. If the authors have adequately addressed your comments raised in a previous round of review and you feel that this manuscript is now acceptable for publication, you may indicate that here to bypass the “Comments to the Author” section, enter your conflict of interest statement in the “Confidential to Editor” section, and submit your "Accept" recommendation.

Reviewer #1: All comments have been addressed

Reviewer #2: All comments have been addressed

2. Is the manuscript technically sound, and do the data support the conclusions?

Reviewer #1: Yes

Reviewer #2: Yes

3. Has the statistical analysis been performed appropriately and rigorously? 

Reviewer #1: Yes

Reviewer #2: Yes

4. Have the authors made all data underlying the findings in their manuscript fully available?

Reviewer #1: Yes

Reviewer #2: Yes

5. Is the manuscript presented in an intelligible fashion and written in standard English?

Reviewer #1: Yes

Reviewer #2: Yes

6. Review Comments to the Author

Reviewer #1: The paper has been further improved and my points were addressed accordingly. Congratulations on this fine piece of work!

Reviewer #2: Thank you very much for addressing all comments! The authors have done an excellent job responding to the questions.

7. PLOS authors have the option to publish the peer review history of their article (what does this mean?). If published, this will include your full peer review and any attached files.

Reviewer #1: Yes: Christian Labenz, MD

Reviewer #2: Yes: Henning Pflugrad, MD

---

## [Editor Report · Acceptance letter]

5 Dec 2019

PONE-D-19-25609R1 

Prediction of overt hepatic encephalopathy by the continuous reaction time method and the portosystemic encephalopathy syndrome test in clinically mentally unimpaired patients with cirrhosis 

Dear Dr. Wernberg:

I am pleased to inform you that your manuscript has been deemed suitable for publication in PLOS ONE. Congratulations! Your manuscript is now with our production department. 

With kind regards,

on behalf of

Dr. Pavel Strnad 

Academic Editor

PLOS ONE